# A Retrospective Study of the Proportion of Women at High and Low Risk of Intrauterine Infection Meeting Sepsis Criteria

**DOI:** 10.3390/microorganisms10010082

**Published:** 2021-12-31

**Authors:** Hen Y. Sela, Vered Seri, Frederic S. Zimmerman, Andrea Cortegiani, Philip D. Levin, Arnon Smueloff, Sharon Einav

**Affiliations:** 1Shaare Zedek Medical Center, Department of Obstetrics and Gynecology, Hebrew University Faculty of Medicine, Jerusalem 9103102, Israel; hysela@szmc.org.il (H.Y.S.); veredsery24@gmail.com (V.S.); asamueloff@szmc.org.il (A.S.); 2Barzilai Medical Center, Department of Intensive Care, Ashkelon 7830604, Israel; 3Department of Surgical, Oncological and Oral Science (Di.Chir.On.S.), University of Palermo, 90133 Palermo, Italy; andrea.cortegiani@unipa.it; 4Department of Anaesthesia, Intensive Care and Emergency, Policlinico Paolo Giaccone, 90127 Palermo, Italy; 5Intensive Care Unit, Shaare Zedek Medical Center, Hebrew University Faculty of Medicine, Jerusalem 9103102, Israel; levinp@szmc.org.il (P.D.L.); einav_s@szmc.org.il (S.E.)

**Keywords:** chorioamnionitis, intrauterine infection, SIRS, sepsis, early identification

## Abstract

The Surviving Sepsis Campaign recently recommended that qSOFA not be used as a single parameter for identification of sepsis. Thus, we evaluated the efficacy of SIRS and qSOFA scores in identifying intrauterine infection. This case–control study evaluates SIRS and qSOFA criteria fulfillment in preterm premature rupture of membranes (*n* = 453)—at high infection risk—versus elective cesarean—at low infection risk (*n* = 2004); secondary outcomes included intrauterine infection and positive culture rates. At admission, 14.8% of the study group and 4.6% of control met SIRS criteria (*p* = 0.001), as did 12.5% and 5.5% on post-operation day (POD) 1 (*p* = 0.001), with no significant differences on POD 0 or 2. Medical records did not suffice for qSOFA calculation. In the study group, more cultures (29.8% versus 1.9%—cervix; 27.4% versus 1.1%—placenta; 7.5% versus 1.7%—blood; *p* = 0.001—all differences) and positive cultures (5.5% versus 3.0%—urine—*p* = 0.008; 4.2% versus 0.2%—cervix—*p* = 0.001; 7.3% versus 0.0%—placenta—*p* = 0.001; 0.9% versus 0.1%—blood—*p* = 0.008) were obtained. Overall, 10.6% of the study group and 0.4% of control met the intrauterine infection criteria (*p* = 0.001). Though a significant difference was noted in SIRS criteria fulfillment in the study group versus control, there was considerable between-group overlap, questioning the utility of SIRS in intrauterine infection diagnosis. Furthermore, the qSOFA scores could not be assessed.

## 1. Introduction

The Surviving Sepsis guidelines mandate the investment of a concerted effort towards the early recognition of severe infection [1]. The use of early warning scores has been associated with earlier treatment and improved mortality in patients with suspected sepsis [2]. Such scores therefore constitute an important part of the assessment of patients with suspected infection. Pregnant women are at increased risk for certain types of infection. In fact, infection is the third most common cause of maternal death [3,4,5]. Yet, the vital signs and laboratory values of pregnant women not only differ from those of the non-pregnant population, but also change with gestational age, which raises questions regarding the value of scores currently used to identify severe infection in the pregnant population.

Several generic scoring systems, including the Modified Early Obstetric Warning System, have been proposed to facilitate early recognition of critical illness, including sepsis, in obstetric patients [6,7,8]. However, these tools are inherently non-specific. The tools used specifically to identify sepsis include the systemic inflammatory response syndromes (SIRS) criteria, sequential organ failure assessment (SOFA) and quick- SOFA (qSOFA) criteria. The SIRS criteria were first recommended for early identification of severe infection [9,10], mostly for the purpose of defining target populations for sepsis studies. In 2016, based on new research, a recommendation was put forward to replace the SIRS criteria with the qSOFA criteria in order to more effectively identify patients outside of the critical care setting who are at high risk of mortality. The qSOFA criteria, which reflect the severity of specific organ failures rather than the immune response to the presence of infection, were proposed to be not only an evidence-based means of recognizing early infection but also clinically more practical [11]. However, recent recommendations have rejected the qSOFA as a single screening tool, due to insufficient sensitivity [12].

Genitourinary infections are very common in pregnant women. One of the more common infections of this type occurring in pregnancy and the peripuerum is intrauterine infection or inflammation (previously known as chorioamnionitis [13]). Preterm premature rupture of membranes (PPROM) is especially associated with an increased risk of intrauterine infection; at least 10% of women with PPROM develop intrauterine infection [14,15,16]. In low risk obstetric populations this risk is much lower and is reported to be between 1.7% and 5% [17,18] Such infections can result in maternal morbidity and mortality and are accompanied by a high rate of fetal loss. Early broad spectrum antibiotic treatment of intrauterine infection may reduce maternal and fetal morbidity and mortality [8,13]. Thus, the early identification of severe infection is vital for reducing the morbidity associated with this condition. This current study investigated the value of various tools, currently and previously recommended for identifying the presence of sepsis in the general population, in identification of intrauterine infection. The primary objective was to evaluate the rate of SIRS and qSOFA criteria fulfillment among peripartum women with a higher likelihood of infection versus those at low risk. The secondary objectives were (a) to evaluate the frequency that the components of these scores are actually available for scoring in the peripartum population and (b) the rate of documented intrauterine infection and the rate of positive cultures (urine, cervix, placenta and blood) in the two groups. 

## 2. Materials and Methods

The study protocol was submitted to the IRB committee and given its retrospective nature, the IRB approved the study with waiver of informed consent (approval number: SZMC-0001-16). Following approval, a retrospective case–control study was conducted on all women who underwent cesarean delivery (CD) during a 10 year period in the Shaare Zedek Medical Center (SZMC), Jerusalem, Israel. 

Clinical setting: The SZMC is a 1000-bed university-affiliated acute care hospital with a Division of Obstetrics that includes a high risk pregnancy unit, two delivery suites with attached dedicated obstetric operating rooms and five maternity wards. The annual rate of admissions for delivery approximated 14,000 during the study period.

Inclusion exclusion criteria: The study group consisted of women admitted to hospital with a diagnosis of PPROM at a gestational age of 24 through 36 weeks, provided they underwent CD within 7 days of admission and prior to week 37. Previous studies have shown that these women have a 10% risk of developing intrauterine infection or inflammation [14]. The control group consisted of subjects undergoing CD at a gestational age of 37 weeks or more, with no rupture of membranes and no trial of labor prior to surgery. In women with these characteristics the risk of intrauterine infection or inflammation is considered approximately 1.7% [17,18], but there is scarce data of good quality on the topic. Women with a vaginal delivery, those with rupture of membranes prior to week 24 and those with rupture of membranes or trial of labor who gave birth in week 37 or later were excluded from both groups. Additionally, women with more than 7 days elapsing from rupture of membranes to delivery were excluded.

Case identification: Relevant cases admitted to the SZMC (August 2005–December 2015) were identified via structured queries to the obstetric electronic medical record (NeSS Technologies, Israel). For the study group, the query first identified all women admitted to the SZMC between August 2005 and December 2015 and then selected those who (1) had undergone CD prior to week 37 and within one week of admission and (2) had an admission or discharge diagnosis of rupture of membranes. If a discrepancy was found between the admission and discharge diagnosis, the medical record was reviewed. In such cases, if rupture of membranes had not been recorded on admission or throughout hospitalization—the case was removed from the study. Most relevant data were available in the NeSS-EMR for the cases identified for this study. However, if required, missing data were also completed from hard copy admission files.

For the control group, the query identified women who had undergone CD at a gestational age of 37–40 weeks with no trial of labor. In this group there was a large quantity of eligible subjects. Therefore, those for whom full data (as described below) was not available via the NESS-EMR were excluded and no review of the hard copy admission files was required.

Variables: The primary outcome measures were (1) the rate of SIRS and qSOFA criteria fulfillment among the study population and (2) the difference in the rates of SIRS and qSOFA criteria fulfillment in the study population as compared to the control population. Secondary outcome measures included (a) the frequency that the components of above scores were actually available the peripartum population and (b) the rate of documented intrauterine infection and positive cultures (urine, cervix, placenta and blood) in the two groups. 

In order to study these outcomes, the data to be collected included the vital signs (heart rate, blood pressure, temperature, respiratory rate) and mental status documented in real time by the nursing staff, as well as complete blood count (CBC), creatinine and bilirubin levels on the day of admission and on POD 0, 1 and 2 and all culture results. 

For purposes of this study, women were considered likely to have intrauterine infection if their discharge notes included a relevant diagnostic code (ICD-9 658.41) or if they had been treated with broad spectrum antibiotics within 24 h of their delivery, these cases were reviewed to ascertain the diagnosis of intrauterine infection. Therefore, the data collected also included the presence of relevant coding and any evidence in the file that the woman had been treated with broad-spectrum antibiotics. During the study years the clinical criteria used for diagnosing intrauterine infection were maternal fever >38 °C along with one or more of the following: maternal or fetal tachycardia, elevated maternal white blood cell count, uterine tenderness and purulent fluid or purulent discharge from the cervical os. We did not use the CDC recommendation of four days of continuous antibiotic usage to identify sepsis, as we did not collect length of treatment because our institutional protocol is to continue antibiotics after CD until either at least 24 h have elapsed without fever or until the return of cultures. In women who deliver vaginally, treatment is continued for at least 24 h post-partum; if there are additional risk factors, antibiotic treatment is continued for at least 24 h afebrile or until return of cultures.

Potential sources of bias: During data collection several issues arose. It was discovered that temperature was not recorded on POD 0 for 88/453 (19.4%) of subjects in the study group and 900/2004 (44.9%) of subjects in the control group. Upon review it was apparent that these omissions were in subjects undergoing an evening operation and arriving in the maternity ward close to midnight. Since these cases comprised a considerable percentage of our study population and the time of omission was consistent, concerns arose regarding potential documentation bias. This was addressed in the analysis by comparing the women with missing data to those without missing data (see Appendix A). The proportion of documentation omissions with regard to blood pressure, heart rate and temperature excluding POD0 was negligible and random, therefore women lacking data regarding any of these were excluded from the study (see Appendix A). An additional major issue that became apparent during data collection was the paucity of documentation of the respiratory status and mental status of women in both the control and the study groups. This deficit precluded calculation of the subjects’ qSOFA scores. 

Method of data collection: The SZMC has separate electronic medical records (EMRs) for obstetrical and all other patient data. During the study period, the obstetric ward was also performing duplicate documentation as changes were being implemented to the EMR for accreditation purposes. Cases were identified via the NESS-EMR, and the vital signs were collected from the NESS-EMR (which was in use in the various obstetric departments). The rest of the data were collected from the hospital-wide EMR. As noted above, when data were unavailable in the EMR, hard copy files were also reviewed to seek additional information. 

For all women, the initial set of data collected included the variables recorded at the time of admission regardless of the location of admission. For women who were hospitalized but did not undergo CD immediately upon admission, data were collected from the period they were observed in the high risk pregnancy unit. Women who remained in the delivery suite more than 24 h before CD had data collected from there. All data collected from the period after CD were taken from maternity ward notes. 

Sample size considerations: The study was designed as a 1:4 case–control study. Based on the data in the literature, we hypothesized that in the study group, the rate of intrauterine infection would be approximately 10% [14,15,16,19], whereas in the control group this rate would be at least 1.7% and no more than 5% [17,18]. According to these data, 437 cases in the study group and 1748 cases in the control group were needed to reject (with a power of 0.95) the null hypothesis that there was an identical rate of fulfillment of sepsis criteria in the study and control groups. Upon initial review of our computerized database, we identified approximately 600 women with a diagnosis of PPROM. We assumed that about 30% of these would be eliminated after manual review. Thus 420 would remain in the study group, of whom approximately 42 (10%) would be found to have an intrauterine infection. No more than 2000 subjects were required in the control group, of whom no more than 100 (5%) would be found to have an intrauterine infection. The probability of a type I error associated with this test of the null hypothesis was calculated as 0.05.

Quantitative variables: The lowest systolic blood pressure and highest pulse and fever recorded each day were used for analysis. The plan was to describe the Glasgow Coma Scale as the total sum rather than its three components. Laboratory data, including leukocyte and platelet count, creatinine and bilirubin, were collected and analyzed as presented by the hospital laboratory.

Data management and statistical analysis: The data were downloaded from the EMR to a Microsoft Excel (Ver. 2010) database and then transferred to SPSS (IBM Corp. Released 2013. IBM SPSS Statistics for Windows, Version 22.0. Armonk, NY, USA, IBM Corp.), which was used for analysis. All data from hard copy patient admission files were manually added to the original file. 

In the first step, descriptive statistics (i.e., numbers, proportion and means) were used to describe the study population as a whole and study and control group characteristics (Table 1).

In the second step, comparisons between women with and without missing data were performed and in the third step, comparisons between the study and control groups were performed. In both of these steps, comparison between proportions was performed using the χ2-score (e.g., for demographic and clinical features) or the Fisher’s exact test (e.g., for the rate of positive cultures). To compare continuous variables the Student’s *t*-test (e.g., for maternal age) or the Mann–Whitney-Wilcoxon test (e.g., for vital signs and obstetric characteristics such as number of gestations and previous CDs) were used. In all tests, two-tailed *p*-values were taken and a *p*-value < 0.05 was considered significant. 

Finally, we performed sensitivity analyses to compare the rate of positive urinary cultures between the two groups, once by assuming that the missing cultures were positive and once by assuming they were negative.

## 3. Results

Of the 142,372 admissions for delivery that took place in the SZMC during the study period, 7499 deliveries occurred at a gestational age of 24–36 weeks. Among these, 621 women were identified as having a presumptive diagnosis of PPROM and also underwent CD. After chart review, 168 were excluded as they did not fulfill the study group criteria in full. Likewise, during the study period 107,401 deliveries took place at a gestational age of 37–40 weeks. Among these, 3159 women underwent CD without a trial of labor. After chart review, 1155 were excluded as they did not meet the overall control group criteria. In total, 453 subjects were included in the study group and 2004 subjects were included in the control group (Figure 1). 

### 3.1. Assessment of Reporting Bias

The results of the comparison of cases with missing data to those with no missing data are presented in Appendix A. In the control group, patients with missing data were slightly younger, had fewer previous pregnancies, had undergone fewer CDs and had a higher incidence of hypertension than those without missing data. No significant differences were observed in the study group.

### 3.2. Description of the Study Population

The average age of the women was 33.6 ± 5.6 years and they had undergone 3.8 ± 2.6 previous deliveries and 1.1 ± 0.9 previous caesarean deliveries. The women in the study group were younger than those in the control group. They were also more likely to be a member of a minority group and were less likely to have completed secondary education. The women in the study group also had fewer deliveries and caesarean deliveries than those in the control group (Table 1).

### 3.3. Comparison between Study and Control Groups—Vital Signs and Complete Blood Count

The women in study group had a higher body temperature than the women in the control group at the time of admission and on POD 0 (36.73 ± 0.4 versus 36.68 ± 0.3, *p* = 0.005; and 36.72 ± 0.5 versus 36.46 ± 0.5, *p* = 0.001, respectively). They also had a higher mean arterial pressures and leukocyte counts throughout admission. Although these differences were all statistically significant, they were clinically meaningless. Women in the study group also had higher heart rates throughout admission, but the statistical significance of this finding varied by day (Table 2). No significant difference in platelet count between the two groups were noted throughout the admission.

### 3.4. Rates of Vital Sign Documentation

The documentation rates of vital sign recordings and complete blood counts were high (>98%) in both groups. No significant difference in documentation rates of these variables was noted between the groups.

### 3.5. Comparison between Study and Control Groups—Fulfillment of SIRS and qSOFA Criteria

Overall, 14.8% (67/453) of the women in the study group and 4.6% (92/2004) the women in the control group fulfilled SIRS criteria at admission (*p* = 0.001). Likewise, on POD1 12.5% (57/453) of the women in the study group and 5.5% (110/2004) of the women in the control group fulfilled SIRS criteria (*p* = 0.001). No significant differences in the rate of fulfillment of SIRS criteria were noted on other hospitalization days (Figure 2). The data in the medical records did not suffice for the calculation of qSOFA scores.

### 3.6. Comparison between Study and Control Groups—Rate of Diagnosis of Intrauterine Infection

Among the study group 48/453 (10.6%) of the women, and 8/2004 (0.4%) of the women among the control group met the criteria used to define the presence of intrauterine infection (*p* = 0.001).

### 3.7. Comparison between Study and Control Groups—Cultures 

#### 3.7.1. Sampling Rates

Urine cultures were obtained from 91.4% (414/453) of women in the study group and 100.0% (2004/2004) of those in the control group (*p* = 0.001). The proportion of women from whom cervical, placental and blood cultures were obtained was considerably lower in both groups. However, significantly more cultures of any kind were obtained from the study group than from the control group—cervix: 135/453 (29.8%) versus 38/2004 (1.9%), placenta: 124/453 (27.4%) versus 22/2004 (1.1%), and blood: 34/453 (7.5%) versus 34/2004 (1.7%), respectively (*p* = 0.001 for all). 

#### 3.7.2. Culture Results

Among the cultures taken, there were significantly more positive cultures in the study group than in the control group; 25/453 (5.5%) versus 60/2004 (3.0%) for urine cultures (*p* = 0.008), 19/453 (4.2%) versus 4/2004 (0.2%) for cervical cultures (*p* = 0.001), 33/453 (7.3%) versus 1/2004 (0.0%) for placental cultures (*p* = 0.001) and 4/453 (0.9%) versus 3/2004 (0.1%) for blood cultures (*p* = 0.008) (Table 3).

#### 3.7.3. Sensitivity Analysis—Urinary Cultures

A sensitivity analysis was only performed for urinary cultures, as these were routinely obtained for most of the study population (whereas other cultures were only obtained when infection was suspected). When the analysis was performed with the assumption that the missing cultures would have been positive, the difference between the study group and the control group increased. When the analysis was performed with the assumption that the missing cultures were negative, the difference between the groups decreased but remained statistically significant (Appendix A).

## 4. Discussion

Sepsis continues to be one of the leading causes of maternal death worldwide, including in the developed world [20]. Late diagnosis and treatment of sepsis has been shown to increase mortality in the general population [21] and an analysis of maternal deaths following sepsis also showed that late detection and treatment contributed to maternal mortality [22]. The risk of sepsis increases in women who undergo caesarean section during labor and those with PPROM [23]. Therefore, early diagnosis of severe acute infection is particularly important in these populations. This retrospective study of 2457 women was designed to examine the proportion of women meeting SIRS and qSOFA criteria among those hospitalized due to PPROM (a population at high risk of intrauterine infection) as compared to the proportion among those at low risk of intrauterine infection. Significantly more women in the high risk group met SIRS criteria than women in the low risk group, especially on admission. However, there were women in both groups that did not meet these criteria despite clear differences in risk and eventual culture positivity. In other words, pregnant/peripartum women with a high likelihood of severe systemic infection may not necessarily fulfill SIRS criteria while those fulfilling SIRS criteria may not really be at risk of systemic infection. These findings are similar to a previous retrospective study, which failed to show an association between SIRS criteria and risk for intensive care unit transfer, sepsis, or death among pregnant women with intrauterine infection [19]. However, unlike the previous study—we were able to evaluate SIRS criteria for the majority of patients included in both arms of the study (Figure 1). In addition, the study group was compared to a control group, and we were able to analyze the applicability of sepsis criteria to multiple points relative to CD.

A second important finding of this study is the extent of missing documentation and cultures in this population. This finding is not unique to our study [19] and offers further support for recent recommendations against the use of qSOFA as a single sepsis screening tool [12]. 

It should be noted that the rate of infection as identified in our study is similar to that of previous studies. Thus, in this study, as in previous studies [14,15,16,19], 10% those with PPROMM developed intrauterine infection. Furthermore, despite the low overall rate of culture collection, the rates of positive urinary, vaginal and placental cultures observed in our study group were similar to those described in previous studies [14,24]. Additionally, our control group had rates of positive urinary cultures similar to those reported in previous studies [25,26,27]. These similarities increase the generalizability of findings to a larger obstetric population.

Our study results support findings from previous studies suggesting that the physiological changes of pregnancy complicate the identification of systemic infection in pregnant women at risk [19,28]. Though there are generally accepted normal ranges in pregnancy for the various components of the SIRS criteria, these are based on very little data. Current definitions of abnormal values in women with PPROM are based on expert consensus only [13]. This study demonstrated a statistically significant between-group difference in the proportion of those meeting SIRS criteria; this difference was maintained at several time points during admission. Nevertheless, because a substantial number of patients in the study group did not meet SIRS criteria, whereas a substantial number of those in the control group did, these criteria do not seem particularly useful for differentiating between women at risk and those not at risk.

As noted above a significant secondary finding is the extent of missing documentation and cultures. This finding may be related to long-standing midwifery traditions [29] and the tendency, as described in other areas [30], to treat the pregnancy rather than the woman carrying it. The qSOFA score was put forward as a tool for identifying sepsis in the general population precisely because its components are used to monitor patients on the ward. However, this assumption does not hold true in obstetric populations. This study and others [29,31] highlight that the relevant maternal parameters (particularly respiratory rate and mental status) are not documented at all in many births. The question arises whether these simple measurements should be incorporated into the obstetric work routine in these departments. 

Despite the similarity between the women in the two groups, more blood, placental and vaginal cultures were taken from those included in the study group. Had these been taken routinely from all women at risk, one could argue that the actual risk profile is associated with a higher rates of positive culture. However, these cultures were not taken routinely, which indicates that the treating physicians had higher suspicion of infection in some cases. These suspicions were later substantiated by an associated higher positive culture rate. In the absence of clear-cut differences between the cases, the question remains as to what led clinicians to suspect infection. If this suspicion can be quantified, it may be possible to establish clearer indicators for a prompt diagnosis.

This study has several limitations. Although this study did include a control arm, it is a retrospective data analysis of an unmatched cohort, and, therefore, its conclusions should be approached with caution. The need for caution is further reinforced by the demographic differences between the study and control groups (Table 1). A further limitation is the lack of full documentation, necessitating the elimination of a substantial percentage of the initial cohort, and making it impossible to evaluate qSOFA at all. In addition, the rate of cultures obtained was lower than expected; still, the percentage of positive cultures was higher in the study group than in the control group, with this difference remaining statistically significant in bi-directional sensitivity analyses for urinary cultures. Furthermore, due to study limitations, we were not able to use the CDC recommendation of 4 days of continuous antibiotic usage to identify sepsis in retrospective studies. Administrative data also has its limitations. Some women included in the study group did not have intrauterine infection but the rate of SIRS criteria fulfillment was still high. The use of administrative data to perform clinical research is a common practice, despite reliance on the accuracy of the codes used. Prior studies have shown that intrauterine infection codes have a negative predictive value of 98% and a positive predictive value of 50% [32]. Finally, there are inflammatory markers that we did not examine. Although several studies have proposed that C-reactive protein (CRP), procalcitonin (PCT) and interleukin 6 (IL6) may be used for early identification of intrauterine infection [33,34], a recent systematic review of diagnostic test accuracy studies noted that there is insufficient evidence to support the use of these markers for diagnosing intrauterine infection in PPROM [35].

## 5. Conclusions

This study found that despite a statistically significant difference in the rate of SIRS in patients with PPROM undergoing CS versus those undergoing elective CS, there is still a considerable overlap between these two groups, thus calling into question the utility of SIRS in the prompt diagnosis of intrauterine infection in pregnant women and possibly in the immediate postpartum period. Additional studies are required to validate our preliminary findings that SIRS criteria should not be used for screening. Missing documentation relevant to the qSOFA score supports recent recommendations against the use of qSOFA as a single sepsis screening tool [12]. This study further emphasizes the need for better maternal monitoring in the peripartum period and a more widespread use of cultures in identifying intrauterine infection in at-risk populations.

## Figures and Tables

**Figure 1 microorganisms-10-00082-f001:**
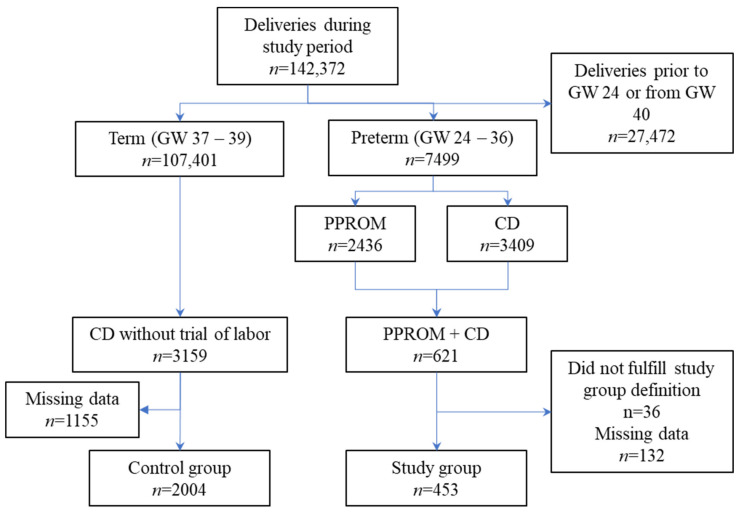
Study inclusion/exclusion process. GW: gestational week; CD: cesarean delivery; PPROM: preterm premature rupture of membranes.

**Figure 2 microorganisms-10-00082-f002:**
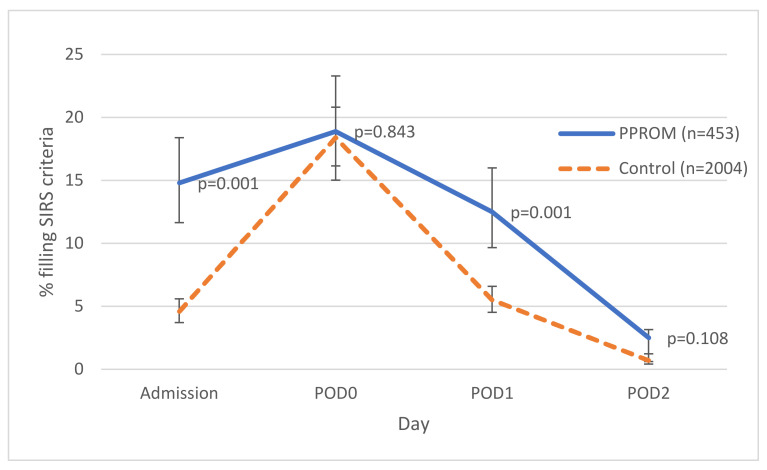
Proportion of patients that fulfilled the systemic inflammatory response syndrome (SIRS) criteria among the study and control groups. POD: post-operative day; PPROM: preterm premature rupture of membranes.

**Table 1 microorganisms-10-00082-t001:** Demographic, obstetric and medical characteristics of study group versus control group. PPROM: preterm premature rupture of membranes; CI: confidence interval.

Characteristic	PPROM *n* = 453 (%)	Control *n* = 2004 (%)	*p*
Maternal age (mean ± CI)	31.5 ± 6.7	33.8 ± 5.3	0.001
>35 year	127 (28.0)	750 (37.4)	0.001
Member of minority	70 (15.5)	220 (11.0)	0.001
Completed secondary education	411 (90.7)	1934 (96.5)	0.005
Gestation number	3.2 ± 2.5	3.9 ± 2.6	0.001
Previous cesareans (mean ± CI)	0.4 ± 0.8	1.1 ± 1.2	0.001
Any previous cesarean	121 (26.7)	1077 (59.9)	0.001
Previous abortion (mean ± CI)	0.7 ± 1.3	0.7 ± 1.1	0.531
In vitro fertilization	126 (27.8)	319 (15.9)	0.001
Twin pregnancy	160 (35.3)	200 (10.0)	0.001
Gestational diabetes	46 (10.2)	269 (13.4)	0.060
Hypertension	18 (4.0)	69 (3.4)	0.581
Hypothyroidism	29 (6.4)	117 (5.8)	0.647

**Table 2 microorganisms-10-00082-t002:** Vital signs and white cell counts of study group versus control group. PPROM: preterm premature rupture of membranes; POD: post-operation day.

Characteristic	DAY	PPROM	*n*	Control	*n*	*p*
Temperature	Admission	36.73 ± 0.4	453	36.68 ± 0.3	2004	0.005
	POD0	36.72 ± 0.5	365	36.46 ± 0.5	1104	0.001
	POD1	36.82 ± 0.5	453	36.81 ± 0.4	2004	0.367
	POD2	36.69 ± 0.4	453	36.65 ± 0.4	2004	0.147
Pulse	Admission	92.7 ± 14.4	453	88.2 ± 11.4	2004	0.001
	POD0	84.8 ± 12.5	453	83.2 ± 10.8	2004	0.06
	POD1	90.6 ± 11.5	453	88.8 ± 9.2	2004	0.029
	POD2	88.9 ± 11	453	87.8 ± 9.6	2004	0.34
Mean arterial pressure	Admission	90.3 ± 10.9	453	85.6 ± 9.3	2004	0.001
	POD0	76.8 ± 10.5	453	74.3 ± 9.5	2004	0.001
	POD1	73.6 ± 9.5	453	70.8 ± 8.2	2004	0.001
	POD2	78.2 ± 10	453	76.8 ± 9.1	2004	0.008
Leukocyte count (×10^3^)	Admission	10.3 ± 9.2	452	9.2 ± 2.2	1997	0.001
	POD0	13.4 ± 11.7	310	11.7 ± 3.3	1617	0.001
	POD1	12.2 ± 11.4	187	11.4 ± 3	587	0.007
	POD2	11.4 ± 9.7	56	9.7 ± 2.6	138	0.005

**Table 3 microorganisms-10-00082-t003:** Microbiological cultures—study group versus control group. PPROM: preterm premature rupture of membranes.

Body Site	Cultures Obtained			Positive Cultures		
	PPROM (*n* = 453)	Control (*n* = 2004)	*p*	PPROM (*n* = 453)	Control (*n* = 2004)	*p*
Urine	414 (91.4)	2004 (100.0)	0.001	25 (5.5)	60 (3.0)	0.008
Cervix	135 (29.8)	38 (1.9)	0.001	19 (4.2)	4 (0.2)	0.001
Placenta	124 (27.4)	22 (1.1)	0.001	33 (7.3)	1 (0.0)	0.001
Blood	34 (7.5)	34 (1.7)	0.001	4 (0.9)	3 (0.1)	0.008

## Data Availability

The data presented in this study are available on request from the corresponding author. The data are not publicly available due to privacy concerns.

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
