# Peer review of "A Retrospective Study of the Proportion of Women at High and Low Risk of Intrauterine Infection Meeting Sepsis Criteria"

_microorganisms, 2021, doi:10.3390/microorganisms10010082_

Round 1
Reviewer 1 Report
I welcome the study performed by Sela et. al. assessing the proportion of women at high and low risk of intrauterine infection meeting SIRS and qSOFA criteria. The authors identified high risk group i.e. patient with PROM who underwent CD and low risk group i.e. patient undergoing CD to determine the proportions of patient meeting SIRS criteria on admission, POD O, 1,2 and 3.
The study and research questions are very appropriate and I applaud the authors on their attempt to answer this important clinical question. All my recommendations are to help strengthen the study.
Abstract:
I will edit the first line to clarify the statement by SSC campaign which suggest not using qsofa as single parameter for identification of sepsis. Small change but a big difference in what the recommendation states.
I will remove qsofa from title as we ended up comparing SIRS criteria only between the 2 groups.
After reviewing the manuscript, I am a bit confused on what the goal of this manuscript is.
Their null hypothesis is that there was an identical rate of intrauterine infection in the 167 study and control groups. The definition of intrauterine infection is based on ICD code and/or use of antibiotics? Yet, there conclusion was there was statistical difference between SIRS criteria between the 2 groups but not enough to use it as a screening tool.
Major Edits
Since, the primary hypothesis is regarding Intrauterine infection, the definition used by the group is ICD code and use of antibiotics.
Is there any reference, which has looked at sens, specificity, PPV and NPV for this ICD code to actually diagnose IU infection? If so, please add the reference
Clarify antibiotic usage. CDC recommends 4 days of continuous antibiotic usage to identify sepsis in retrospective study. https://www.cdc.gov/sepsis/pdfs/Sepsis-Surveillance-Toolkit-Mar-2018_508.pdf.
What was the usual duration of antibiotic for this group?
If the question is regarding the utility of SIRS criteria for screening for sepsis, I would suggest the author to also check the number of patient identified with IU infection in each group, checking SIRS criteria 24 hour prior to start of antibiotics and doing sensitivity/specificity/PPV/NPV. As the diagnosis of sepsis is still in identifying if infection is the cause of SIRS response.
3rd, meeting SIRS criteria on the day of OR is perfectly appropriate so excluding POD O is appropriate.
Expand on IU infection result description, as this was the null hypothesis.
It is interesting the POD day 2 SIRS score to be similar between the group. Looking at the graph closely close to 1% and 2.5% of patient meet SIRS criteria on day 2 in each group. Suggesting that we should not use SIRS for screening based on such small % of patient meeting criteria is a overstatement.
Reviewer 2 Report
This study is proposing additional scientific support for other already published papers focusing on insufficient SIRS or qSOFA score in relation to gravidity, which is important. However, I have some major recommendations.
- Usage of "qSOFA" in the title as well as in key words needs to be reconsidered as it is not supported with any relevant data in this study. Authors only show Figure 2, which mentions SIRS criteria only. I would suggest then omitting the usage of qSOFA if it is lacking.
- IRB approval - this needs to be explained, not only shortening should be used.
- Figure 2 - the legend of this figure is poorly explained, therefore the figure is not understandable for the readers.
- Were the methods used for intrauterine infection determination united? Were the other measurements performed similarly (blood pressure, creatinine, bilirubine)? In fact, the values of creatinine and bilirubin are not stated in the submitted paper. They need to be added.
- Line 300 says there are studies of PPROMM and 10% of them developed infection. Although authors cite only one study.
- Table 1 - are all information needed and necessary to show in this study?
- There exist more modern and fast techniques beside the use of cultures that should be mentioned in this study. I suggest adding them, discussing them.
Round 2
Reviewer 1 Report
The authors have appropriately addressed all my concerns.
Reviewer 2 Report
I have no other comments